# TEMPO: Time Series Understanding via Discrete Tokenization

**Riccardo Maggioni** [1]

## Abstract

Time-series signals are a primary data modality in medicine, manufacturing, and engineering, yet integrating them into large language models remains an open challenge. Prior approaches either project continuous encoder representations into the LLM via cross-attention, which can collapse during training, or serialize individual timestep amplitudes as text, discarding temporal structure. We propose TEMPO, which extends an LLM's vocabulary with discrete codes from a context-aware transformer tokenizer using finite scalar quantization. Signal and text tokens share the same embedding space and self-attention stream, requiring no architectural modification and fewer than 1% trainable parameters. On five benchmarks spanning activity recognition, sleep staging, bearing-fault diagnosis, and time-series QA, TEMPO with a 4B-parameter backbone matches or exceeds cross-attention and text-serialization baselines while producing natural-language reasoning grounded in the signal.

## 1. Introduction

Foundation models have transformed language and vision, but structured data, including tabular records and time series, remains largely served by per-dataset models. The core difficulty for time series is representation: how should a raw waveform enter an LLM so that the model can reason over it jointly with text? This question is central to building foundation models for structured data that transfer across heterogeneous datasets and domains, rather than training one model per task.

To see why the problem is hard, consider a vibration trace from a gearbox showing a sharp spectral peak. The pattern alone is ambiguous: it could indicate incipient bearing damage, a benign cold-start transient, or manufacturing variance from a recently replaced part. Disambiguation requires the operating context, the maintenance history, and domain knowledge of how each cause manifests; the output is necessarily free-form text combining signal evidence, context, and reasoning, not a label.

Following the simplification arc in vision–language models, from gated cross-attention (Alayrac et al., 2022) to discrete visual tokens sharing the text vocabulary (Meta, 2024), we propose TEMPO, which extends the LLM vocabulary with discrete codes from a context-aware transformer tokenizer using finite scalar quantization (FSQ; (Mentzer et al., 2024)). Signal and text tokens interleave in a single flat sequence processed by standard self-attention, with no cross-attention, no gating, and no architectural changes to the backbone.

**Contributions.**

1. *Discrete vocabulary extension for time-series reasoning.* Signal tokens join the LLM vocabulary with no architectural changes, enabling joint signal–text reasoning across three sensor domains.

2. *Context-aware discrete tokenization.* A transformer-based FSQ tokenizer produces codes that depend on the full input signal, not just local patches. The 625-code codebook achieves 100% utilization.

3. *Competitive accuracy with natural-language output.* TEMPO reaches 86–89% across three tasks while generating step-by-step reasoning grounded in the signal, a capability closed-set classifiers cannot provide.

## 2. Related Work

**Time-series foundation models.** Most TSFMs target forecasting or representation learning. Chronos (Ansari et al., 2024) quantizes scalars into uniform bins for a T5 backbone; TimesFM (Das et al., 2024) pretrains a decoder-only transformer for probabilistic forecasting; Moirai (Woo et al., 2024) introduces any-variate attention; Mantis (Feofanov et al., 2025) learns contrastive embeddings for classification. None produce natural-language reasoning; their outputs are forecasts, labels, or embeddings.

[1]Forgis. Correspondence to: Riccardo Maggioni <riccardo.maggioni@forgis.com>.

*Proceedings of the $43^{rd}$ International Conference on Machine Learning*, Seoul, South Korea. PMLR 306, 2026. Copyright 2026 by the author(s).

**Signal-to-LLM integration.** ChatTS (Xie et al., 2025), OpenTSLM (Langer et al., 2025), TsLLM (Parker et al., 2025), and ITFormer (Wang et al., 2025) project continuous encoder representations into the LLM as soft prompts or via cross-attention. Chat-TS (Quinlan et al., 2025) discretizes individual timestep amplitudes into fixed bins. TokenCast (Tao et al., 2025) trains a CNN-based VQ-VAE tokenizer for patch-level discrete codes used in forecasting. Outside time series, vocabulary extension with discrete tokens has been applied to vision (Meta, 2024) and audio (Nguyen et al., 2025).

TEMPO differs on two axes. First, the tokenizer is context-aware: a transformer encoder applies self-attention over the full input before quantization, so a local pattern maps to different codes depending on surrounding context, unlike CNN-based tokenizers limited to a fixed receptive field or scalar binning operating on individual timesteps. Second, FSQ replaces VQ-VAE quantization, eliminating codebook collapse; our 625-code codebook achieves 100% utilization.

## 3. Method

Three components implement TEMPO (Figure 1): a context-aware signal tokenizer, vocabulary extension, and a two-stage training recipe. We describe each below.

**Preprocessing.** Each signal channel is standardized per-channel using its own mean $m_c$ and standard deviation $s_c$: $\hat{x}_{t,c} = (x_{t,c} - m_c)/s_c$. This disentangles two kinds of information. The signal tokens encode *temporal structure* in a scale-invariant form, letting a single codebook serve signals from any domain or sensor calibration. *Scale* is preserved through a separate channel: $(m_c, s_c)$ are appended to the textual context, from which any absolute amplitude can be recovered. The LLM can then reason about magnitudes when they matter, such as vibration severity thresholds or clinical voltage ranges, while the tokenizer itself remains scale-invariant. This decomposition mirrors how analysts read signals: structural interpretation from the waveform, quantitative assessment from the numbers.

**Context-aware tokenization.** The tokenizer is a 4-layer transformer encoder followed by FSQ. Non-overlapping patches of $L=4$ timesteps are linearly projected to $d=128$ dimensions, processed by global self-attention, then projected to $d_{\text{fsq}}=4$ dimensions. Each dimension is quantized to one of $K=5$ levels, yielding $5^4=625$ implicit codebook entries:

$$c_i = \text{FSQ}\big(\text{TransformerEnc}(\hat{x}_{1:T})_i\big), \quad c_i \in \{0, \ldots, 624\}. \quad (1)$$

A symmetric transformer decoder reconstructs the signal from quantized codes. The tokenizer is pretrained on $\sim$500K windows from 9 domains (forecasting, UCR, bear-ing, ECG, financial, IMU, synthetic) with MSE and multi-scale spectral loss on a single A10 GPU in 4 hours, then frozen.

Context-awareness distinguishes this design from prior tokenizers: because the encoder attends over the full signal before quantization, the same local waveform produces different codes depending on surrounding context. A peak during a rising trend encodes differently from the same peak on a flat baseline.

**Vocabulary extension and LLM integration.** We add 627 tokens: 625 codes `<ts_0>`–`<ts_624>` plus delimiters `<ts_start>` and `<ts_end>`. New embeddings are initialized at the mean norm of existing embeddings. Signal and text tokens form a single interleaved sequence:

```
[user] Signal:  (mean=0.12, std=0.45)
<ts_start> <ts_207> <ts_14> ...  <ts_end>
What fault does this bearing have?
[assistant]
```

**Training.** Training proceeds in two stages. **Stage 0** (embedding alignment): only the 627 new embeddings are trained on $\sim$200K samples: 50% discriminative tasks (22 synthetic signal types paired with 19 task types, including binary questions, identification, captioning, comparison, and counting, with 20% noise augmentation), 20% code imputation (masked span prediction), and 30% M4 captions (signal-to-text grounding). The goal is to teach the embeddings *what signals look like*. **Stage 1** (multi-task fine-tuning): DoRA adapters (Liu et al., 2024) (rank $r=32$, applied to all attention and MLP projections) are unfrozen alongside the signal embeddings. The training mixture ($\sim$265K samples) covers classification, anomaly detection, temporal reasoning, chain-of-thought diagnosis, and open-ended description across 12 time-series domains, with $\sim$50K text-only instruction samples to prevent catastrophic forgetting. Training uses DDP on $8\times$H100 GPUs; total cost from pretrained LLM to working system is under $50.

## 4. Experiments

### 4.1. In-Distribution Evaluation

We fine-tune the Stage 1 checkpoint on domain-specific training splits and evaluate on held-out test sets. All experiments use Qwen3-4B as the backbone.

TEMPO reaches 89% on HAR, 85% on sleep staging, and 86% on bearing-fault diagnosis (Table 1), exceeding all cross-attention and soft-prompt baselines by 15–36 points and approaching task-specific classifiers. On TSAQA anomaly detection, TEMPO matches the best 8B text-serialization model (91.3%) with a 4B backbone. Do-

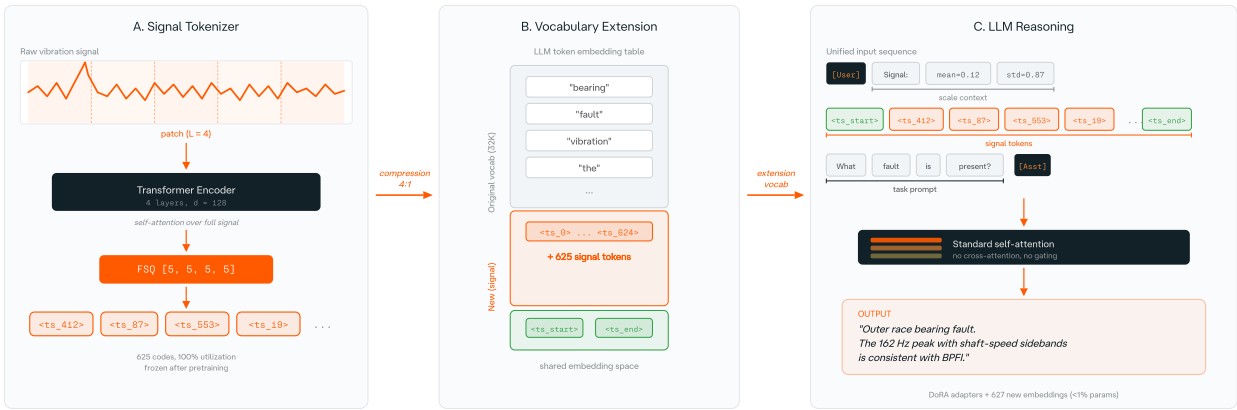

*Figure 1.* TEMPO architecture. A frozen FSQ-Transformer tokenizer (4:1 compression) maps each 4-timestep patch to one of 625 codes via self-attention over the full signal. Codes become tokens in the LLM vocabulary; only DoRA adapters and 627 new embeddings are trained (<1% of parameters).

*Table 1.* Comparison across specialized classifiers, text-serialization LLMs, and TEMPO on five benchmarks. HAR, Sleep, Bearing: macro F1 (%). TSAQA: accuracy (%). UCR: mean F1 on a 10-dataset subset.

| Category | Method | HAR | Sleep | Bearing | TSAQA cls/ad |
|---|---|---|---|---|---|
| *Specialized classifiers (no rationale)* | | | | | |
| Feature | RF | 96 | 63 | 96.1 | — |
| E2E | AttnSleep | — | 84.4 | — | — |
| E2E | WDCNN | — | — | 99.2 | — |
| *LLMs (text serialization)* | | | | | |
| Zero-shot | GPT-4o | 2.95 | 15.5 | 36.6 | 28.8/55.7 |
| Tuned | LLaMA3.1-8B | — | — | — | **91.3**/91.0 |
| *LLMs (cross-attention / soft prompt)* | | | | | |
| SoftPrompt | Llama-1B | 65.4 | 69.9 | — | — |
| Flamingo | Llama-1B | 62.9 | 49.3 | — | — |
| *TEMPO (ours, discrete tokenization)* | | | | | |
| FSQ-T | Qwen3-4B | **89.0** | **85.0** | **86.0** | 80.5/**91.3** |

*Table 2.* Out-of-distribution evaluation on Time-MQA. None of these questions were seen during training. Published baselines (†) from (Kong et al., 2025).

| Model | Params | Anom. | Cls. | Judg. |
|---|---|---|---|---|
| GPT-4o[†] | >200B | 64 | 32 | 72 |
| Qwen-2.5-7B[†] | 7B | 68 | 52 | 82 |
| TEMPO (1-shot) | 1.7B | 56 | 62 | 52 |

(52%) lags, likely reflecting the smaller backbone's limited reasoning capacity.

### 4.3. Cross-Domain Code Sharing

A useful discrete vocabulary should capture generic temporal primitives that transfer across domains. We tokenize held-out windows from 6 domains and measure pairwise cosine similarity of per-domain code activation vectors. The FSQ-Transformer achieves 100% codebook utilization with uniformity $H/H_{\max}$=0.889. Cross-domain sharing is structured: ECG and UCR share codes (cosine = 0.78) via quasi-periodic transients; financial and forecasting data cluster separately (cosine = 0.73) via trend primitives. All pairwise similarities exceed a permutation null by >44$\sigma$.

### 4.4. Qualitative: Context-Dependent Reasoning

Classification accuracy alone cannot capture TEMPO's central claim: that discrete tokenization enables reasoning conditioned jointly on signal and text. We evaluate this on CWRU bearing-fault data (SKF 6205-2RS, 1800 RPM) using a single Qwen3-4B checkpoint fine-tuned on 20K bearing-fault samples.

main specialists (RF at 96%, WDCNN at 99.2%) remain superior on narrow tasks, but these are purpose-built classifiers that cannot produce reasoning or generalize across domains.

### 4.2. Out-of-Distribution Evaluation

To test generalization, we evaluate on 600 questions from Time-MQA (Kong et al., 2025), an external benchmark not seen during training, using a smaller Qwen3-1.7B backbone.

With one-shot prompting, the 1.7B TEMPO reaches 62% on classification (vs. 52% for Qwen-2.5-7B fine-tuned on Time-MQA and 32% for GPT-4o zero-shot) and 56% on anomaly detection (Table 2), despite being 4× smaller and receiving no Time-MQA training data. Judgment accuracy

*Table 3.* Severity-graded recommendations on CWRU bearing data. All severities correctly classified; action escalates with ISO zone.

| Diameter | RMS | ISO | Class | Action |
|----------|-----|-----|-------|--------|
| 0.007" | 0.29 | A | outer_race | *No action* |
| 0.014" | 0.20 | B | outer_race | *Monitor* |
| 0.021" | 0.49 | C | outer_race | *Plan replacement* |
| 0.028" | 0.87 | D | outer_race | *Replace now* |

**Context-dependent interpretation.**  We present the same vibration signal under two textual contexts: an aged bearing where incipient faults are expected, and a newly installed non-OEM part during commissioning. The model diagnoses outer-race fault in the first case; in the second, it questions whether the spectral features reflect a genuine fault or manufacturing variance from the non-OEM part, demonstrating that the LLM conditions its interpretation on both the signal tokens and the accompanying textual metadata.

**Severity-graded recommendations.**  Across four fault severities (Table 3), the model correctly classifies all as outer-race fault while escalating recommendations from "no immediate action" to "schedule replacement immediately" based on the ISO vibration zone provided in context.

**Signal-conditioned vs. post-hoc reasoning.**  To illustrate the structural limitation of post-hoc explanation, we pair a *deliberately incorrect* classifier label (inner-race fault) with the raw signal serialized as numerical text and pass both to GPT-4o for explanation. Despite receiving the full signal, GPT-4o produces a fluent rationalization of the wrong diagnosis, citing BPFI harmonics and progressive RMS trends as supporting evidence, without flagging any contradiction. In contrast, TEMPO, conditioned on discrete signal tokens, correctly identifies outer-race fault. This suggests that text-serialized signals do not provide LLMs with sufficient grounding to override a confident but incorrect prior.

## 5. Discussion and Conclusion

**Context-dependent codes as learned preprocessing.** Traditional time-series pipelines rely on hand-crafted feature engineering: spectral analysis, envelope detection, statistical tests. The FSQ-Transformer's context-dependent coding achieves an analogous function automatically. When identical local patches receive different codes depending on the surrounding signal regime, the tokenizer is implicitly computing context-relative features that would otherwise require explicit, domain-specific design.

**Cross-domain code sharing enables transfer.**  The full 625-code vocabulary is activated across the pretraining mixture, with inter-domain sharing far exceeding chance ($44\sigma$ above a permutation null; Section 4.3). Sharing

is *structured*: ECG and UCR signals share codes via quasi-periodic transients, while financial and forecasting data cluster via shared trend primitives. This suggests the codebook learns a taxonomy of temporal primitives, not domain-specific shapes, supporting the transfer-across-heterogeneous-datasets paradigm central to foundation models for structured data.

**Limitations.**

1. *Compression ratio.* The 4:1 compression ($L{=}4$ timesteps per code) limits temporal resolution. Signals requiring sub-patch precision, such as QRS morphology in ECG at low sampling rates, may lose clinically relevant detail. Adaptive patch sizes or hierarchical tokenization could address this.

2. *Multivariate handling.*  Multi-channel signals are currently tokenized per-channel and concatenated in the token sequence, allowing the LLM to perform cross-channel reasoning via self-attention. For high-dimensional sensor arrays ($>$10 channels), this produces long sequences that may exceed context limits. Addressing this while keeping sensor fusion inside the LLM remains an open problem.

3. *Tokenizer coverage.* The FSQ-Transformer is pretrained on 500K windows from 9 domains. Signals from represented domains or domains sharing similar temporal primitives can be handled without retraining; however, fundamentally different signal families may produce poorly distributed codes, and domain-adaptive tokenizer fine-tuning is unexplored.

**Future work.**  Because signal tokens share the same vocabulary and attention stream as text, the LLM can *generate* time-series codes as part of its output. This opens modalities beyond classification and QA: autoregressive forecasting via future code generation, counterfactual reasoning ("what if the fault worsened?") via conditional code synthesis, and imputation via masked code infilling, all within a single architecture with no decoder head swap.

**Conclusion.**  TEMPO integrates time-series reasoning into pretrained LLMs through discrete vocabulary extension, with no architectural modification and fewer than 1% of parameters trained. It approaches task-specific classifiers on accuracy while producing natural-language reasoning grounded in signal evidence and textual context, capabilities structurally unavailable to label-only classifiers or post-hoc explanation pipelines. Total training cost is under $50.

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

## A. Embedding Quality: Linear Probe

To assess whether the tokenizer–LLM interface preserves discriminative information, we extract time-series embeddings after Stage 0 alignment, mean-pool across code positions, and train a logistic regression classifier on 15 UCR datasets. All three configurations use Qwen3-1.7B, trained for 10 epochs on the same alignment dataset (200K samples).

*Table 4.* Linear probe accuracy (logistic regression on mean-pooled TS embeddings) after Stage 0 alignment.

| Tokenizer | Avg Probe Acc (15 UCR) | Val Loss |
|---|---|---|
| FSQ-T (non-RoPE) | **73.5%** | 1.42 |
| CNN+VQ | 71.4% | 0.94 |
| FSQ-T-RoPE | 69.6% | 1.97 |

The FSQ-Transformer with absolute position embeddings achieves the highest probe accuracy (73.5%), outperforming both CNN+VQ (+2.1 pp) and RoPE (+3.9 pp). Probe accuracy does not correlate with validation loss: CNN+VQ has the lowest val loss but intermediate probe accuracy, suggesting val loss measures reconstruction quality while probe accuracy measures *discriminative* embedding quality, which is more relevant for downstream tasks.

**Per-dataset breakdown.** Table 5 reports test accuracy on each of the 15 UCR datasets.

*Table 5.* Per-dataset linear probe accuracy (%). Bold = best per dataset.

| Dataset | FSQ-T | CNN-VQ | FSQ-T-RoPE |
|---|---|---|---|
| ArrowHead | 57.7 | **66.9** | 55.4 |
| CBF | 54.9 | **60.2** | 51.7 |
| ECG200 | **75.0** | 69.0 | 66.0 |
| FaceFour | **61.4** | 51.1 | 53.4 |
| GunPoint | **92.7** | 75.3 | 86.7 |
| Trace | **100.0** | 94.0 | 92.0 |
| Wafer | **93.7** | 92.4 | 94.7 |
| SyntheticControl | **55.3** | 51.7 | 39.0 |
| TwoPatterns | **61.7** | 59.5 | 63.2 |
| FaceAll | **42.8** | 41.1 | 39.2 |
| StarLightCurves | 92.1 | 85.9 | **92.8** |
| Chinatown | 87.5 | **86.6** | 81.1 |
| ItalyPowerDemand | 83.1 | 74.6 | **84.0** |
| SmoothSubspace | 53.3 | **69.3** | 58.7 |
| UMD | 91.0 | **93.1** | 86.1 |
| **Average** | **73.5** | 71.4 | 69.6 |

FSQ-T wins on 9/15 datasets, with the largest gains on datasets with clear morphological structure: Trace (100% vs. 94/92%), GunPoint (92.7% vs. 75.3/86.7%), ECG200 (75% vs. 69/66%), i.e. datasets where context-dependent coding is most beneficial. CNN-VQ outperforms on shape-outline datasets (ArrowHead, UMD, SmoothSubspace) where local shape is sufficient. RoPE underperforms its non-RoPE variant, suggesting rotary embeddings may interfere with fixed-length signal structure where absolute position carries semantic meaning.

## B. Training Details

### B.1. Model Architecture

The TS tokenizer is **frozen** during all training phases. Only the 625 new token embeddings and the DoRA adapters receive gradients.

### B.2. Training Phases

**Stage 0 — Embedding Alignment.**

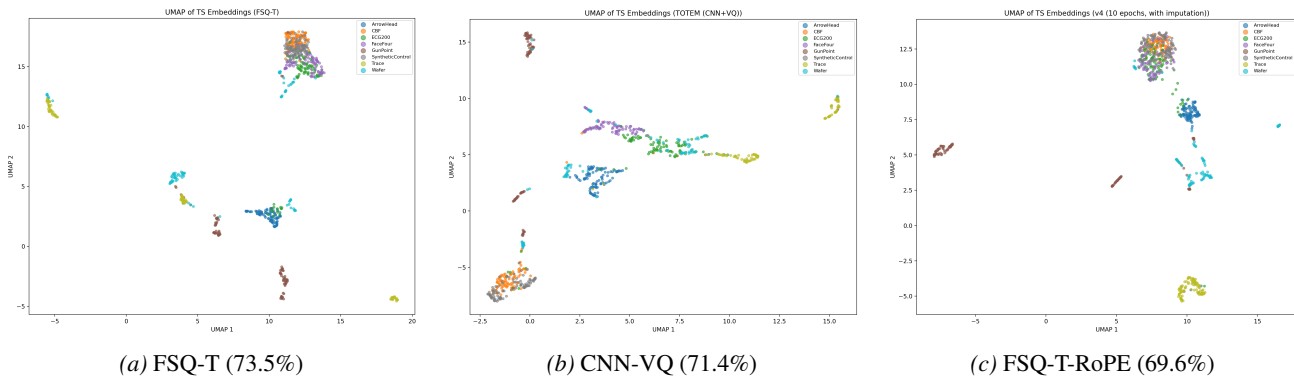

*(a)* FSQ-T (73.5%)      *(b)* CNN-VQ (71.4%)      *(c)* FSQ-T-RoPE (69.6%)

*Figure 2.* UMAP of mean-pooled TS embeddings after Stage 0 alignment, colored by UCR dataset. FSQ-T produces the tightest, most separated clusters.

*Table 6.* Model configurations.

|  | 1.7B | 4B |
|---|---|---|
| Backbone | Qwen3-1.7B | Qwen3-4B |
| Hidden dim | 2048 | 3584 |
| Layers | 28 | 36 |
| Attention heads | 16 | 28 |
| Vocab size (base) | 151,643 | 151,643 |
| Vocab size (extended) | +625 TS tokens | +625 TS tokens |
| *Adapter (DoRA)* | | |
| Rank ($r$) | 32 | 32 |
| Alpha ($\alpha$) | 64 | 64 |
| Dropout | 0.05 | 0.05 |
| Target modules | q, k, v, o, gate, up, down | |
| Trainable params | $<1\%$ | $<1\%$ |
| *Time-series tokenizer* | | |
| Codebook size | 625 | 625 |
| Levels (FSQ) | $[5, 5, 5, 5]$ | $[5, 5, 5, 5]$ |
| Compression | 4:1 | 4:1 |
| Patch size | 4 | 4 |
| Max TS tokens | 750 | 750 |
| Max sequence length | 2048 | 2048 |

**Stage 0 dataset.** 200K samples: 50% discriminative tasks (22 signal types, 19 task types, 20% noise augmentation), 20% code imputation (masked span prediction), 30% M4 captions (signal-to-text grounding).

**Stage 1 — Multi-Task Fine-Tuning.**

**Stage 1 dataset.** 267K samples: TSAQA-Benchmark (139K), HAR-CoT (20K), Sleep-CoT (9K), CWRU Bearing-CoT (20K), ECG-QA-CoT (20K), text instructions (50K SlimOrca + Alpaca).

### B.3. Tokenizer Training

### B.4. Infrastructure and Cost

## C. DoRA Rank Ablation

We evaluate three DoRA ranks on multi-task bearing fault diagnosis (186K samples, Qwen3-4B). At $r = 64$, the model achieves perfect classification but suffers catastrophic forgetting (outputs only the bearing template). At $r = 16$, capacity is insufficient (9/30 errors). $r = 32$ provides the best trade-off: 86% accuracy with full conversational ability.

*Table 7.* Stage 0 hyperparameters.

| Parameter | Value |
|---|---|
| Learning rate | $1 \times 10^{-3}$ |
| Scheduler | Linear warmup + linear decay (min $0.1\times$) |
| Warmup fraction | 10% of total steps |
| Epochs | 5 |
| Batch size (per GPU) | 8 |
| Gradient accumulation | 2 |
| Effective batch size | 128 (8 GPUs) |
| Sequence packing | ON (pack_length = 1024) |
| Optimizer | AdamW ($\beta_1 = 0.9$, $\beta_2 = 0.999$) |
| Weight decay | 0.01 |
| Precision | BF16 mixed precision |
| Parallelism | DDP ($8\times$ H100 80GB) |

*Table 8.* Stage 1 hyperparameters.

| Parameter | Value |
|---|---|
| Learning rate | $2 \times 10^{-5}$ |
| Scheduler | Linear warmup + linear decay (min $0.1\times$) |
| Warmup fraction | 10% of total steps |
| Epochs | 3 |
| Batch size (per GPU) | 8 |
| Gradient accumulation | 2 |
| Effective batch size | 128 (8 GPUs) |
| Precision | BF16 mixed precision |
| Parallelism | DDP ($8\times$ H100 80GB) |

*Table 9.* FSQ-Transformer tokenizer training.

| Parameter | Value |
|---|---|
| Architecture | Transformer enc (4L) + FSQ + Transformer dec (4L) |
| $d_{\text{model}}$ | 128 |
| Attention heads | 4 |
| FSQ levels | $[5, 5, 5, 5]$ (625 codes) |
| Patch size | 4 timesteps |
| Input length | 256 |
| Training data | 500K windows from 9 domains |
| Loss | MSE + multi-scale spectral + temporal smoothness |
| Optimizer | AdamW, lr = $10^{-3}$ |
| Hardware | $1\times$ A10G (24GB), $\sim$4 hours |
| Codebook utilization | 100% (625/625) |
| Uniformity ($H/H_{\text{max}}$) | 0.889 |

*Table 10.* Compute cost breakdown.

| Stage | Hardware | Wall time | Est. cost |
|---|---|---|---|
| Tokenizer training | $1\times$ A10G | $\sim$4 hr | $\sim$\$3 |
| Stage 0 (1.7B) | $8\times$ H100 | $\sim$5 hr | $\sim$\$49 |
| Stage 1 (1.7B) | $8\times$ H100 | $\sim$3.5 hr | $\sim$\$34 |
| Evaluation (per run) | $1\times$ A10G | $\sim$20 min | $\sim$\$0.25 |
| **Total (1.7B)** | | $\sim$13 hr | $\sim$**\$86** |

