# OpenReview forum: "TEMPO: Time Series Understanding via Discrete Tokenization"
_ICML.cc/2026/Workshop/FMSD — FMSD @ ICML 2026 Poster_

### Official Review · Reviewer_abiz · 2026-05-21
**Discrete Tokenization for Time Series Language Alignment**

**Rating:** 7
**Confidence:** 3

**Review:**

The paper introduces a time series codebook and an efficient training pipeline to align time series data with text. Although the method itself is not entirely novel, this type of discrete tokenization approach has not been widely explored in the time series domain. The results show that the model is able to capture and contextualize nuances in time series by aligning them with language representations. Overall, the idea is sound, the experiments are well designed, and some desirable emerging properties arise from this approach.
That said, further experimentation could strengthen the work, particularly on tasks such as contextualized or reasoned forecasting, anomaly detection, and imputation. It would also be interesting to compare this general approach more directly against domain specific foundation models for time series, for example comparing TEMPO against models such as MantisV2 on classification tasks. Additionally, experiments that corrupt either the text context or the time series inputs could help disentangle how much the model benefits from the textual information versus the numerical signal itself during classification. Such analyses would provide deeper insight into what the model is actually learning and where the gains are coming from.

---

### Official Review · Reviewer_33Z6 · 2026-05-21
**Promising discrete tokenization approach for time-series.**

**Rating:** 6
**Confidence:** 3

**Review:**

Summary:
The paper introduces TEMPO, a method of integrating time-series signals into LLMs by converting time-series windows into discrete tokens using a context-aware FSQ transformer tokenizer. These tokens are added to the LLM vocabulary allowing time series waveform tokens and text tokens to be processed in the same embedding space without major architectural modifications (practices like cross-attention or gating).

Strengths:
- The paper is well motivated and fits the workshop theme.
- Figure 1 explains the architecture cleanly.
- the use of FSQ over VQ-VAE is well motivated
- out-of-distribution evaluation results suggest promising directions.

Areas for Improvement:
- Signal grounded claim made in the abstract is mainly supported by qualitative examples
- The baseline comparison does not cleanly isolate the contribution of TEMPO’s signal interface (Table 1 is a mix of different methods  making it difficult to specifically attribute gains specifically to the FSQ-T method).
- The paper makes claims that a transformer encoder lets a local pattern map to different codes, however this is not directly evaluated.
- Outside of downstream task performance, the tokenizer's information preservation is not characterized. FSQ code utilization is mentioned, but there is not an analysis of metrics such a reconstruction quality, spectral preservation, or failure cases.
-- Appendix A provides useful linear-probe evidence but it is still indirect evidence.


Detailed Comments:
- The paper could benefit from deployment-facing metrics such as inference latency, memory usage, or throughput to better support claims of efficiency/parameter-efficiency (to further support the fewer than 1% trainable parameters claim).
- The paper could benefit from ablations where parameters are fixed and only the signal interface/tokenization method is changed
- More tests to quantitatively show that the natural-language reasoning is grounded in the signal tokens (signal shuffling for example).
- Cross-domain code sharing is reported, but more analysis on how that can be further optimized through methods like pruning would be an interesting direction to explore future directions/ limitations.


Justification of Score:
- Paper is relevant, clearly written, and presents a promising interface for time-series LLMs.
- Some claims are backed qualitatively, more experimentation needed for signal preservation. More controlled experiments would better validate the method.